# ARMesh: Autoregressive Mesh Generation via Next-Level-of-Detail Prediction

**Jiabao Lei**[1*], **Kewei Shi**[2*], **Zhihao Liang**[3], **Kui Jia**[1†]

[1] The Chinese University of Hong Kong, Shenzhen
[2] The University of Hong Kong
[3] Tencent Hunyuan

https://jblei.site/proj/armesh

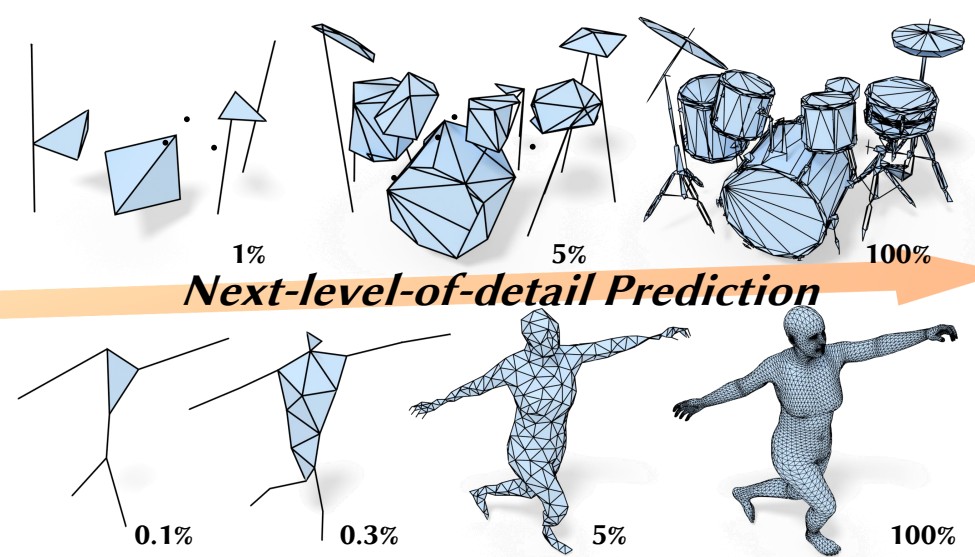

## Abstract

Directly generating 3D meshes, the default representation for 3D shapes in the graphics industry, using auto-regressive (AR) models has become popular these days, thanks to their sharpness, compactness in the generated results, and ability to represent various types of surfaces. However, AR mesh generative models typically construct meshes face by face in lexicographic order, which does not effectively capture the underlying geometry in a manner consistent with human perception. Inspired by 2D models that progressively refine images, such as the prevailing next-scale prediction AR models, we propose generating meshes auto-regressively in a progressive coarse-to-fine manner. Specifically, we view mesh simplification algorithms, which gradually merge mesh faces to build simpler meshes, as a natural fine-to-coarse process. Therefore, we generalize meshes to simplicial complexes and develop a transformer-based AR model to approximate the reverse process of simplification in the order of level of detail, constructing meshes initially from a single point and gradually adding geometric details through local remeshing, where the topology is not predefined and is alterable. Our experiments show that this novel progressive mesh generation approach not only provides intuitive control over generation quality and time consumption by early stopping the auto-regressive process but also enables applications such as mesh refinement and editing.

_____________________
[*] Equal Contribution; [†] Corresponding Author: *kuijia@cuhk.edu.cn*

39th Conference on Neural Information Processing Systems (NeurIPS 2025).

# 1   Introduction

3D triangular surface mesh is the predominant representation in graphics. Due to its explicit control nature, compactness, and strong compatibility with modern graphics pipelines, it is important and has been widely used in various applications, such as the gaming industry and film production. Since 3D meshes are discrete and often non-watertight, autoregressive (AR) models are increasingly popular for directly generating meshes instead of relying on intermediate representations like implicit fields, which may struggle with non-watertight, non-manifold, or thin-sheet surfaces. While AR models prove highly effective in modeling discrete ordered sequences in many other domains, such as natural languages, their application to meshes is less straightforward since mesh vertices and faces are inherently unordered. Prior works typically assign lexicographic orders [29] (for instance, from left to right and bottom to top in a canonical coordinate system) or face traversal orders [4, 48, 60] to transform a mesh into a 1D sequence, and later employ an autoregressive model to learn the sequence directly for mesh generation. However, these artificially created orders are often less meaningful in geometry, failing to capture the overall shape during generation, and do not align well with how humans perceive objects in a hierarchical, coarse-to-fine manner.

Such an issue also exists for AR models in 2D, where a raster scanning order for images is easy to implement but not optimal. Recent approaches, such as VAR [49], instead define the AR sequence as predicting the next scales, *i.e.*, generating a higher-resolution image from a lower-resolution one. We are therefore prompted to explore *what the "next-scale" prediction for 3D meshes could be.*

Unlike images or voxel grids that are structured regularly and allow for easy up- or down-sampling to obtain different resolutions, the "scale," more commonly known as level-of-detail (LOD) in graphics, is rarely explored for 3D meshes with deep learning due to the irregular nature and complexity of explicitly managing a mesh's connectivity and topology. Nevertheless, one can still define the LOD for a mesh in terms of the number of faces or vertices. The widely accepted mesh simplification algorithms, such as QSlim [16], help by progressively reducing the face count of a mesh through the repeated collapsing of edges, where an edge connecting two vertices is merged into a single vertex, thus creating different levels of detail for the mesh. By leveraging a mesh simplification algorithm, we are motivated to train an AR model to learn from this reversed simplification sequence and generate meshes by progressively adding geometric details through local remeshing.

Our paper primarily draws upon two works [17, 41] in the traditional area of mesh processing from several decades ago. Specifically, we propose an adapted version of [17], termed GSlim in this paper, which is designed to operate on and simplify simplicial complexes (introduced later in Section 3.1), a generalization of triangular meshes, within a unified framework. The algorithm is enhanced with the capability to allow for topological changes during simplification. With our algorithm, a mesh, regardless of how complex its initial topology may be, will ultimately be reduced to a single point, benefiting from the use of simplicial complexes[2], which is unsupported by QSlim. The algorithm can provide information as a byproduct on how we simplify a mesh in the LOD order. By reversing the simplification process, we can obtain a 1D refinement sequence that progressively refines a mesh through local remeshing, an idea that has been roughly discussed by Popović and Hoppe [41]. The above algorithms essentially convert a mesh into a 1D sequence. We then design a tokenization scheme to discretize the 1D sequence into compact tokens and subsequently train an autoregressive (AR) transformer using the tokenized sequences. This process can also be referred to as next-level-of-detail prediction, as the model essentially predicts refinement operations that add details to construct the next LOD. By employing our method, we can easily obtain a mesh with a user-specified LOD/complexity by early stopping the AR generation, which was unachievable in earlier work. Our experiments also show that such a trained transformer model is a useful and effective shape encoder that can faithfully capture shapes with high fidelity in the task of unconditional mesh generation.

In summary, to achieve direct mesh generation in the order of LODs, this paper proposes reversing the mesh simplification algorithm, thereby converting a mesh into a coarse-to-fine mesh refinement sequence. We train an AR model on these sequences to predict the next LODs. During generation, the mesh initially starts from a single point and is iteratively refined according to the predictions made by the AR model, progressively adding geometric and topological details through local remeshing. Experiments show its streaming generation with flexible controllability over quality and time usage.

---

[2]If we do not use simplicial complexes, an isolated single point cannot be described along with a triangular mesh. Additionally, simplicial complexes accommodate various types of meshes, including those with different topologies, non-watertight or non-manifold cases, mixed dimensions, and so on, as indicated by [17].

## 2 Related Work

**Indirect Mesh Generation.** A mesh can be indirectly obtained by converting it from other representations. Researchers have shifted their focus to studying generation with implicit fields after recognizing the ease of using these fields to handle different topologies [6, 34, 39]. Numerous studies [7, 8, 13, 15, 25, 26, 27, 28, 39, 43] have emerged along this line. Some aim to describe shapes purely geometrically, typically using signed distance fields [6, 34, 39], while others further explore how to encode both geometry and appearance within the same representation, typically employing neural radiance fields [20, 24, 35, 36, 55]. Their representations can be either continuous [34, 35, 36, 39] or discrete [28, 30, 43]. They often rely on a post-processing step of iso-surface extraction [31] to convert their representation into a surface mesh. The drawbacks of indirectly generating meshes prompt us to study direct mesh generation methods, as detailed below.

**Direct Mesh Generation.** Initial attempts in this area involve deforming a shape template (*e.g.*, a sphere) to a target [54, 58], which is limited to a fixed topology. Some approaches attempt to explicitly modify the topology [38] or leverage local surface patches to adapt the topology [18, 19], but still fail to produce a mesh with concise tessellation. Some methods capable of obtaining a concise tessellation are based on intermediate representations and need to derive the mesh indirectly from them [8, 13]. PolyGen [37] is the first to directly generate a mesh by explicitly generating its vertices and faces. MeshGPT [45] learns codebooks to facilitate the encoding of triangular faces; some methods [3, 4, 21, 29, 60] strive to develop efficient tokenization strategies that convert a mesh into a one-dimensional sequence; some researchers [21, 53] study the architecture for efficient training of a mesh generation network. However, although there have been numerous works in this area, the paradigm they use mainly originates from [37], where the mesh is constructed face by face. In contrast, we attempt to generate a mesh progressively in the order of levels of detail.

**Levels-of-Detail.** Representing signals with multiple resolutions can generally be effective. Traditional methods [10, 16, 22, 42] focus on designing simplification algorithms that produce multiple levels from a given mesh input. Some works in deep learning [5, 36, 40, 47] suggest producing implicit fields hierarchically or using hierarchical data structures to facilitate fine-grained representation; however, they do not natively operate on the mesh itself. This is mainly due to the challenges associated with handling a mesh's connectivity. A recent work [48] in direct mesh generation attempted to incorporate an additional token for face count control; however, it does not intrinsically model different levels of detail, still essentially treating various levels as distinct meshes. A concurrent work named VertexRegen [61] is built upon progressive meshes [22], which require the initialization to be homeomorphic to the target mesh. However, our method does not have these restrictions.

## 3 Preliminaries

### 3.1 Generalizing Mesh into Simplical Complex

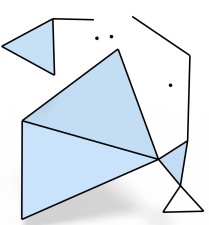

A mesh can be considered exclusively composed of triangles. However, triangles alone are insufficient in this paper. Our method relies on a new representation called simplicial complex (SC), which is a generalization of meshes. A simplicial complex further generalizes a mesh to accommodate (isolated) points and line segments that are not necessarily part of any triangles. Figure 2 illustrates this idea. We sometimes refer to an SC simply as a mesh in this paper for simplicity. Another benefit of using SC is its conciseness and abstractness in describing geometry. For example, line segments are better suited for describing skeletons, while an SC composed exclusively of points is essentially a point cloud.

Figure 2: An simplical complex example.

### 3.2 QSlim: The Standard Quadric-Based Simplification Algorithm

To make our paper self-contained, we briefly introduce the widely accepted mesh simplification algorithm QSlim [16] here before formally introducing the GSlim algorithm that we actually used. For a triangular face $\sigma$ defined by the plane $\mathbf{n}^\top(\mathbf{x} - \mathbf{p}) = 0$ with $\|\mathbf{n}\| = 1$ and $\mathbf{p}$ being the barycenter, we can easily see that the squared distance from a point $\mathbf{x} \in \mathbb{R}^3$ to the plane $\mathbf{n}^\top(\mathbf{x} - \mathbf{p}) = 0$ can be calculated as:

$$Q_\sigma(\mathbf{x}) = \left(\mathbf{n}^\top(\mathbf{x} - \mathbf{p})\right)^2 = \mathbf{x}^\top \mathbf{A}\mathbf{x} + 2\mathbf{b}^\top \mathbf{x} + c$$
$$\text{with} \quad \mathbf{A} = \mathbf{n}\mathbf{n}^\top, \quad \mathbf{b} = -\mathbf{A}\mathbf{p}, \quad c = \mathbf{p}^\top \mathbf{A}\mathbf{p}$$

The above distance, also called the cost, is uniquely defined by the triplet $Q_\sigma = (\mathbf{A}, \mathbf{b}, c)$, which is referred to as the fundamental quadric of $\sigma$. We note that since geometric error can be measured in total as the sum of costs, the quadric is essentially additive. The quadric of a vertex $v$ is therefore the sum of the quadrics of all faces incident to it, that is, $Q_v = \sum_{\sigma \in \mathcal{N}_v} Q_\sigma$, where $\mathcal{N}_v$ is the set of faces incident to $v$. The quadric of an edge $e$ is defined as the sum of the quadrics of its two endpoints, that is, $Q_e = Q_{v_1} + Q_{v_2}$, where $v_1$ and $v_2$ are the two endpoints of $e$. If the edge $e$ will collapse, it will merge $v_1$ and $v_2$ into a single vertex $v^*$ which is the minimizer of $Q_e$, and the quadric of $v^*$ is updated with $Q_{v^*} = Q_{v_1} + Q_{v_2}$. For simplicity, $v^*$ is practically determined by

$$v^* = \underset{u \in \{v_1, v_2, \frac{v_1 + v_2}{2}\}}{\arg\min} Q_e(u)$$

with the collapse cost being $Q_e(v^*)$. This means the collapse position is taken from either the endpoints or the midpoint of the edge. We can calculate the collapsing costs for every edge in the mesh and employ a greedy strategy to iteratively collapse the edge with the minimum cost found. A min-heap data structure will help find the minimum. The algorithm terminates when certain criteria are met, such as when the face count falls below the desired number. At the end of the algorithm, we will obtain a coarse mesh that is homeomorphic (has the same topology) to the original mesh.

To preserve boundary edges, the algorithm needs to generate a perpendicular plane that runs through the edge. These constraint planes are later converted into quadrics, weighted with a large penalty, and added to the initial quadrics for the endpoints of the edge. Apart from the inconvenience of preserving boundaries, there are also several limitations, including unchangeable topology and the inability to handle isolated points and edges. These limitations make the QSlim algorithm unusable for our task, since we envision that any mesh will eventually be simplified to a single vertex, which will require changes to both the topology and simplex dimension during simplification.

## 4 Method

### 4.1 GSlim: Generalized Mesh Simplification Algorithm

The GSlim algorithm we used addressed the aforementioned limitations (Section 3.2). It is adapted from [17] to enable topological changes. It is compatible with SC and supports reducing a mesh to the coarsest shape possible, which is a single point that also serves as the starting point for generation.

Unlike QSlim, which only defines fundamental quadrics for triangles, GSlim defines them for every simplex[3]. The quadric coefficient $\mathbf{A}$ for a simplex $\sigma$ of intrinsic dimension $d$ is instead calculated as

$$\mathbf{A} = \mathbf{I} - \sum_{i=1}^{d} \mathbf{e}_i \mathbf{e}_i^\top$$

where $\{\mathbf{e}_i\}$ is its tangent orthonormal basis of $\sigma$, and $\mathbf{I}$ is an identity matrix.
A larger quadric value in the above formulation will indicate that updates involving $\sigma$ are difficult, thereby postponing its edge collapse. Similarly, we need to aggregate nearby quadrics to compute the edge collapse quadric $Q_e$, the calculation process of which has been detailed in Section 3.2. During the aggregation, we can optionally weight the fundamental quadrics with a large penalty factor to adjust the strength of preserving that simplex. For example, if we assign a large penalty to the boundary edges, it will attempt to preserve the boundary, which is much easier than QSlim. We have included an ablation study in Section 5.1 to investigate which set of penalty factors is the best.

It is interesting to note that for a mesh with $n$ vertices, the algorithm requires $n - 1$ steps to simplify it until only a single vertex remains, as each edge collapse reduces the vertex count by exactly one.

### 4.2 PSC: Reversing Simplification into Generation

Since the simplification process is a many-to-one mapping, it is not inherently reversible unless additional information is recorded to specify how to reverse this process. The reverse operation of edge collapse is called "vertex split," and an example is provided in Figure 4. By reversing simplification with vertex splits, one can reformulate a simplicial complex into a new representation called a Progressive Simplicial Complex (PSC) [41], which consists of a starting point followed by a sequence of vertex splits for coarse-to-fine refinement. Since in PSC only new simplices are added to the existing mesh with new indices, without altering the indices of pre-existing simplices, to convert an edge collapse sequence into a vertex split sequence, one must also reorder the simplices' indices so that they are in ascending order. The following information must be recorded for each vertex split:

---

[3]A simplex may refer to a point ($d = 0$), an edge ($d = 1$), or a triangle ($d = 2$).

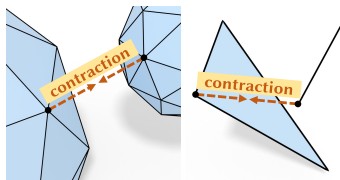

Figure 3: Two examples of generalized edge collapse. Two points will collapse into one point after contraction, leading to topological changes.

Before the algorithm begins, we need to specify a set of candidate vertex pairs for collapse. Unlike the convention [16] that uses candidate pairs from triangles, we further include "virtual edges" that bridge up disconnected components to facilitate topological changes. Figure 3 presents two examples of virtual edges for collapse. Following [41], these virtual edges are derived from the Delaunay tetrahedralization [1, 57] of the input vertices. After the simplification is applied, the algorithm will produce a sequence of increasingly coarsened meshes as side products, visualized from right to left in Teaser. In the next section, we will discuss what information should be recorded during progressive simplification to facilitate our future use.

- **which vertex to split**: an integer index is sufficient for selecting the vertex for splitting
- **how the split is performed**: a boolean indicating whether the current position remains unchanged or becomes the midpoint of the split vertex pair
- **the relative positional offset**: an offset vector for translating the new vertex position relative to the current vertex position
- **how the topology changes in adjacency**: a list of topological labels for adjacent simplices

A vertex split operation can be represented as a JSON configuration, which may look like:

```
{ "vsid":0, "midpoint":true, "offset":[1.2,0.7,2.5], "topo":[1,2,5,3,3,4,6,7] }
```

The first three are quite straightforward, while the fourth is less intuitive, and we will elaborate on what it actually means and how we derive it by explaining the example below.

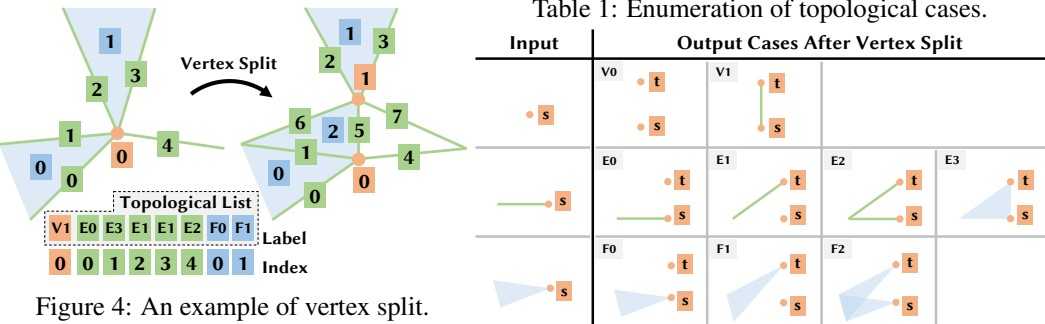

Figure 4: An example of vertex split.

Table 1: Enumeration of topological cases.

In Figure 4, a vertex **0** is split into two, resulting in the creation of a new vertex **1**. Since the vertex split only influences its adjacent primitives (vertex **0** itself, five edges **0–4**, and two faces **0,1**), the topological changes must occur solely in its adjacency. We can use a list called "topological list" to carry the topological change information for each adjacent simplices (always including vertex **0** itself), and its length is exactly the number of adjacent primitives, as shown at the bottom of Figure 4. The list is first sorted by the dimension of the primitives, and then by ascending indices. Each entry in the list represents a classification label ranging from $0 \sim 8$, corresponding to the nine possible cases (`V0`, `V1`, `E0`, `E1`, `E2`, `E3`, `F0`, `F1` and `F2`) summarized in Table 1.

Let's interpret the topological cases in Table 1. For a single vertex marked as **s** (source), it can either be split to form a new vertex **t** (target) or create a new edge **st** connecting **s** and **t**. They are respectively denoted as case `V0` and `V1`. Similarly, there are four cases for an edge: `E0` merely adds a **t**; `E1` switches the edge to connect with **t**; `E2` creates a new edge to connect with **t**; and finally, `E3` creates a new triangle. The explanation for triangle cases is analogous and omitted for brevity.

With Table 1, we can interpret the case presented in Figure 4. For instance, for edge **1** in Figure 4, after splitting vertex **0** to create vertex **1**, a new face **2** is created, corresponding to case `E3` in Table 1. We note that creating face **2** also implies that all its sub-simplices exist or are created as well, including edges **6**, **5**, and **1**. This also suggests a constraint on the topological labels, *i.e.*, case `V0` is inherently incompatible with `E3`, because the former suggests that edge **st** (specifically, the edge **5**) exists, while the latter does not. There are many more hard constraints like this that the configuration must adhere to. The original paper [41] provides only incomplete rules and cannot be used directly for our purposes. We analyze and summarize all topological constraints into the following four rules:

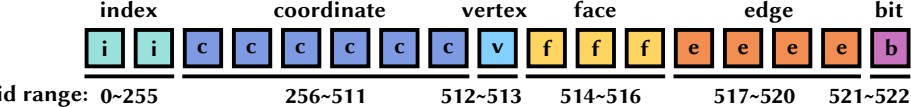

Figure 5: An example of a tokenization layout for a refinement operation.

1. If V0, then cannot have E3.
2. If $v \subset e_1, e_2$ and $e_1, e_2 \subset$ F0, then $e_1, e_2 \neq$ E1.
3. If $v \subset e_1, e_2$ and $e_1, e_2 \subset$ F1, then $e_1, e_2 \neq$ E0.
4. If $v \subset e_1, e_2$ and $e_1, e_2 \subset$ F2, then $e_1, e_2 \neq$ E0 and $\neq$ E1.

A detailed explanation of rules (2-4) is provided in the footnote[4]. These four rules are complete and can be rigorously derived by thoroughly enumerating all possible cases in a table and summarizing the rules that encompass each entry. They also indicate that we can modify only the predictions of the edges to ensure that all the topological label predictions are consistent. We will leverage this property to design our tokenization layout, as shown in Figure 5, where the edge predictions follow the vertex and face predictions to facilitate the constrained decoding process introduced in Section 4.3.

By formulating the vertex split operations (refinement sequence) as a series of JSON configurations, we are able to progressively reconstruct a simplicial complex from a single point with the sequence. In the next section, we will detail how to turn this into a learnable process with AR models.

### 4.3 Auto-Regressive Learning from Refinement Sequences

After obtaining the refinement sequences, we can utilize transformers to learn from these sequences[5]. We need to tokenize the sequence before feeding it into the transformer network. Unlike [56], which directly converts the structured input to textual format, we organize it in a compact layout. We show the operation layout after tokenization in Figure 5. In this example, there are three faces and four edges incident to the selected vertex, resulting in three squares colored in yellow and four squares colored in coral. We now explain the layout in detail: (1) The first two bytes encode the vertex index using int16, and each slot ranges from 0 to 255; (2) The subsequent six bytes encode the relative offset using $3 \times$ fp16, with the first three bytes storing their most significant bytes, and each slot ranges from 256 to 511; (3) We then store the topological labels for the selected vertex, as well as incident faces and edges, respectively. There are only two cases for the selected vertex, so it ranges from 512 to 513. There are only three cases for the incident faces, so it ranges from 514 to 516. There are four cases for the incident edges, so it ranges from 517 to 520; (4) The last byte denotes a boolean indicating whether the current vertex is the midpoint after the vertex split. After all vertex split operations are tokenized, we should be able to obtain a long sequence by concatenating all tokens for autoregressive learning. Figure 6 below illustrates the network design for learning.

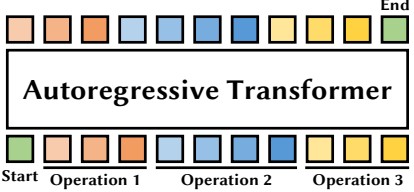

Figure 6: An example of autoregressive learning of three refinement operations.

We simply adopt a transformer network [52] to fit the sequences. Since we primarily focus on unconditional generation in this paper, the network receives no input conditions. The network is responsible for predicting the next token until it reaches the end token and stops generation. However, predictions generated by the transformer may not always correspond to a valid operation, as analyzed in Section 4.2. We may need a solution to determine whether the prediction is valid, possibly through constrained decoding, as shown below, to address this issue and ensure validity.

Apart from the four rules summarized in Section 4.2, there are additional constraints, such as those related to the vertex index and the relationships among different refinement configurations. To facilitate implementation, we can manually hardcode a function, denoted as $\phi(x_i \mid x_1, \ldots, x_{i-1})$, that takes the newly generated token $x_i$ as input, conditioned on all the previous tokens $\{x_1, \ldots, x_{i-1}\}$, and outputs a boolean value indicating whether the new token $x_i$ is compatible with the previous

---

[4]The second rule states that if there are two edges $e_1$ and $e_2$ incident to the split vertex and are subset to a face with a topological label of F0, then the topological labels of $e_1$ and $e_2$ cannot be E1. The explanation for the other rules is quite similar.

[5]We would like to note that our major contribution (Sections 4.1 and 4.2) is theoretically independent of how the learning scheme is designed; we simply present the simplest learning solution here.

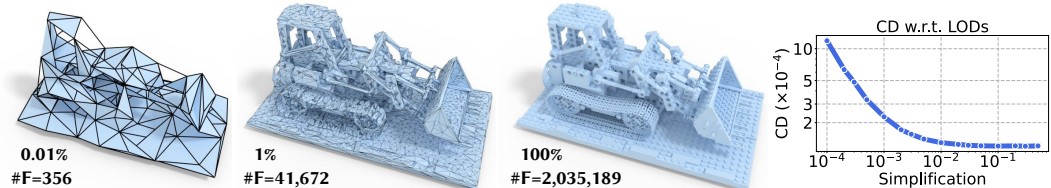

Figure 7: Mesh simplification results at different levels of detail. Our method incorporates isolated points and edges, which also help to approximate the original fine shapes. When simplifying to $1\%$, the geometric accuracy remains nearly the same.

tokens $\{x_1, \ldots, x_{i-1}\}$. We hardcode the function $\phi$ in Python with caching. The basic idea to ensure validity is to use a depth-first search-based tree traversal combined with random sampling to introduce diversity. The function $\phi$ is evaluated at each node, and the most probable node is selected for deeper traversal. If it cannot proceed further, it backtracks to the parent nodes to find alternative solutions. Using this constrained decoding strategy provides an implementation guarantee that the operations predicted by the transformer are always consistent with the previously generated ones.

# 5 Experiments

**Dataset.** Our 3D mesh data mainly come from Stanford 3D Scans [50], Thingi10K [63], NeRF [35], AMASS [33], and ShapeNet [2]. The ShapeNet data has been preprocessed by Siddiqui et al. [45], which contains $< 1,700$ vertices and $< 800$ faces for each mesh, and we follow their settings.

**Metrics.** Following previous works [32, 45, 51, 62], we adopt the following metrics for evaluation: Minimum Matching Distance (MMD), Coverage (COV), and 1-Nearest-Neighbor Accuracy (1-NNA). For MMD, lower values are better; for COV, higher values are better; and for 1-NNA, 50% is optimal. These metrics are calculated based on Chamfer Distance (CD). To measure visual similarity, we also use FID and KID scores to assess visual quality, with lower values being better. We also report mesh compactness as the average number of vertices and faces per mesh.

**Implementation Details.** For simplification, we set the penalty factors for vertices, boundary edges, and faces to 0, 1, and 1, respectively, as their default values in our experiments. We follow the standard techniques in LLM, employing Byte Pair Encoding (BPE) to compress tokens. The entire vocabulary size is 16,384, resulting in a length reduction of $2 \sim 3$ times. We basically follow the setup of MeshGPT [45] for our comparative experiments. We adopt a 12-layer transformer with a width of 768 for training, utilizing variable-length bf16 flash attention [11, 12] for efficient training. The average batch size is $\approx 60$ per GPU. The learning rate is $10^{-4}$ for pre-training across all categories and $10^{-5}$ for supervised fine-tuning on specific categories (chair, table, bench, and lamp). The model is pretrained on 4 H20 GPUs for $\approx 4$ days and fine-tuned on 2 H20 GPUs for $\approx 2$ days.

## 5.1 Ablation Studies

**Simplification with Different Levels-of-detail.** In this part, we study how the geometric reconstruction accuracy varies w.r.t. different LODs using our simplification algorithm. We test a very complex LEGO shape from [35] that contains up to 2 million faces. Visualization results, along with the numerical accuracy, are presented in Figure 7. We find that our simplification algorithm indeed produces meshes with different LODs that consistently mimic the original shape.

**Simplification on Non-manifold Meshes.** We study the necessity of using our proposed generalized simplification (GSlim) algorithm over the widely used QSlim [16] and present the results in Figure 8. Since QSlim cannot handle non-manifold cases, it either crashes during processing or produces erroneous results. Our algorithm is often useful since many datasets contain non-manifold cases; using GSlim will enable direct processing of a non-manifold mesh without preprocessing.

**Simplification with Different Penalties.** To investigate how the penalty factors affect the simplification results, we conduct experiments and present the visualizations in Figure 9. We find that without the boundary edge penalty applied, the simplified mesh cannot maintain a hole in the middle, and the leg length is clearly shortened. With the vertex penalty applied, the resulting triangles become close

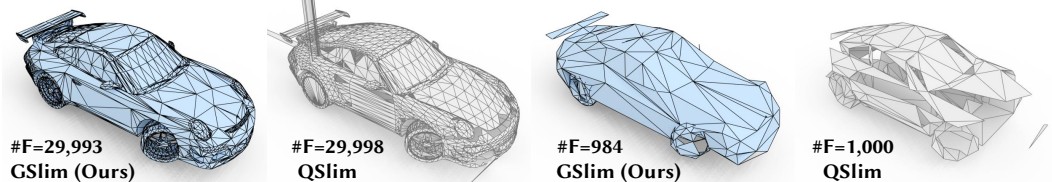

Figure 8: Different mesh simplification algorithms. The popular QSlim algorithm may result in broken and cracked surfaces, while our algorithm (GSlim) still captures the overall shape perfectly, even when the mesh is non-manifold. The QSlim we used is implemented by MeshLab [9].

to equilateral; however, it cannot adaptively distribute its triangles to fit the geometric curvature. The best configuration is therefore $\text{VEF} = (0, 1, 1)$, which is the default for all our other experiments.

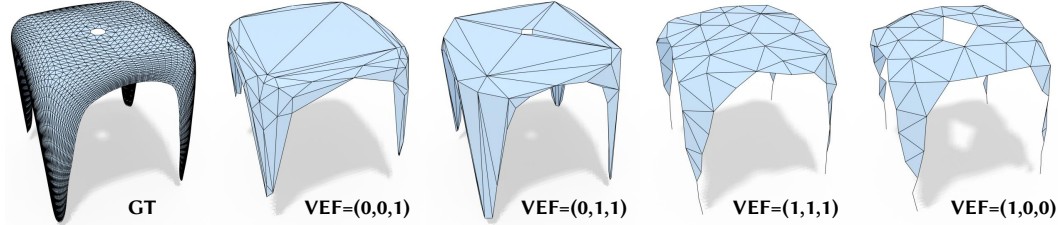

Figure 9: Different penalty factors used in simplification. The four right subfigures show the results with only $2\%$ refinement steps applied. The notation "VEF" at the bottom denotes the penalty values for vertices, boundary edges, and faces, respectively. It is clear that $\text{VEF} = (0, 1, 1)$ is the best.

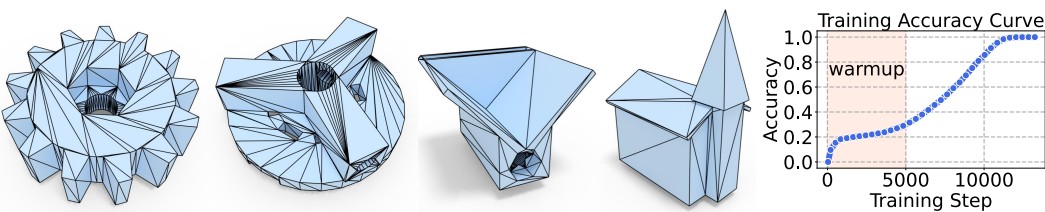

Figure 10: Mesh collection learning with 100 objects taken from the Thingi10K [63] dataset.

## 5.2 Direct Mesh Encoding

**Direct Single Mesh Encoding at Different Levels of Detail.** In this part, we study and compare methods capable of obtaining meshes with different resolutions in the context of single mesh fitting. From Figure 11, we find that our method can faithfully reconstruct the shape, while other methods, such as DMesh [46], require significantly more training steps to achieve better results. Additionally, EdgeRunner [48], a pretrained model that receives a point cloud as input, produces only three LODs and struggles to accurately reconstruct shapes at higher LODs. These three LODs not only may correspond to poor quality and deviate significantly from the claimed number of generated faces but are also insufficient to meet practical requirements due to the limited choices of LODs. In contrast, our method can produce any LODs as needed, and the number of generated vertices is precise.

**Direct Mesh Collection Encoding with Transformer Networks.** Following [44], we investigate whether the refinement sequences can be effectively learned by the transformer using 100 objects. Although simple, the learning process could fail if the sequences are meaningless (nearly random) or if local minima exist, which makes convergence unstable and slow. We present some fitted meshes from the Thingi10K [63] dataset, along with the accuracy curve shown in Figure 10. Fortunately, we did not observe any of these obstacles during training. After the learning rate warms up (increasing linearly to $10^{-4}$), we find that our method converges quickly to a very high accuracy, indicating its effectiveness in learning the refinement sequences with transformer networks.

## 5.3 Unconditional Mesh Generation

We conducted unconditional generation experiments on four categories of the ShapeNet [2] dataset. Visualization results are presented in Figure 12. PolyGen [37] tends to produce simple and incomplete

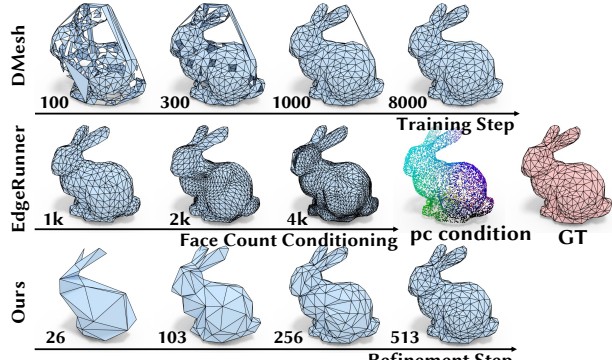

Figure 11: Different LOD methods in mesh reconstruction. EdgeRunner [48] is conditioned on point clouds.

Table 2: Quantitative results of unconditional mesh generation for two categories of the ShapeNet [2] dataset with different LOD ratios.

| Class | Ratio | COV↑ | MMD↓ | 1-NNA | FID↓ | KID↓ | |V|/|F| |
|-------|-------|------|------|-------|------|------|---------|
| Lamp | 0.1 | 57.37 | 2.76 | 72.47 | 49.91 | 0.0246 | 23 / 36 |
| | 0.2 | 65.45 | 2.05 | 63.21 | 27.91 | 0.0096 | 44 / 80 |
| | 0.5 | **71.79** | 1.54 | **51.74** | 10.44 | 0.0013 | 109 / 209 |
| | 1.0 | 70.68 | **1.54** | 41.20 | **6.60** | **0.0005** | 216 / 418 |
| Bench | 0.1 | 41.07 | 1.67 | 86.53 | 41.80 | 0.0187 | 22 / 30 |
| | 0.2 | 47.40 | 1.15 | 83.83 | 21.94 | 0.0106 | 42 / 74 |
| | 0.5 | 54.84 | 0.87 | 66.88 | 10.23 | 0.0022 | 105 / 196 |
| | 1.0 | **56.29** | **0.75** | **39.81** | **2.63** | **0.0001** | 208 / 384 |

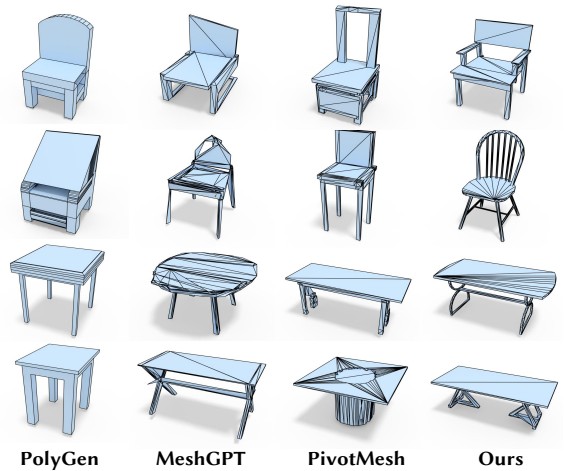

Figure 12: Visualization of unconditional direct mesh generation on the ShapeNet [2] dataset.

Table 3: Quantitative results on unconditional generation across four categories of the ShapeNet [2] dataset. Numerics are copied from [45].

| Class | Method | COV↑ | MMD↓ | 1-NNA | FID↓ | KID↓ | |V|/|F| |
|-------|--------|------|------|-------|------|------|---------|
| Chair | AtlasNet [19] | 9.03 | 4.05 | 95.13 | 170.71 | 0.169 | 2500 / 4050 |
| | BSPNet [8] | 16.48 | 3.62 | 91.75 | 46.73 | 0.030 | 673 / 1165 |
| | Polygen [37] | 31.22 | 4.41 | 93.56 | 61.10 | 0.043 | 248 / 603 |
| | GET3D [15] | 40.85 | 3.56 | 83.04 | 81.45 | 0.054 | 13725 / 27457 |
| | GET3D* | 38.75 | 3.57 | 84.07 | 78.29 | 0.065 | 199 / 399 |
| | MeshGPT | **43.28** | 3.29 | 75.51 | 18.46 | 0.010 | 125 / 228 |
| | Ours | 36.67 | **2.44** | **67.40** | **1.54** | **0.0001** | 198 / 377 |
| Table | AtlasNet [19] | 7.16 | 3.85 | 96.30 | 161.38 | 0.150 | 2500 / 4050 |
| | BSPNet [8] | 16.83 | 3.14 | 93.58 | 30.78 | 0.017 | 420 / 699 |
| | Polygen [37] | 32.99 | 3.00 | 88.65 | 38.53 | 0.029 | 147 / 454 |
| | GET3D [15] | 41.70 | 2.78 | 85.54 | 93.93 | 0.076 | 13767 / 27537 |
| | GET3D* | 37.95 | 2.85 | 81.93 | 50.46 | 0.037 | 199 / 399 |
| | MeshGPT | **45.68** | 2.36 | 72.88 | 6.24 | 0.002 | 99 / 187 |
| | Ours | 32.94 | **1.94** | **69.46** | **2.05** | **0.0002** | 156 / 305 |
| Bench | AtlasNet [19] | 20.53 | 2.47 | 90.58 | 189.39 | 0.163 | 2500 / 4050 |
| | BSPNet [8] | 28.74 | 2.05 | 88.44 | 59.11 | 0.030 | 457 / 756 |
| | Polygen [37] | 51.92 | 1.97 | 76.98 | 49.34 | 0.031 | 172 / 430 |
| | MeshGPT | 55.23 | 1.44 | 68.24 | 8.72 | 0.001 | 159 / 291 |
| | Ours | **56.29** | **0.75** | **39.81** | **2.63** | **0.0001** | 208 / 384 |
| Lamp | AtlasNet [19] | 19.97 | 4.68 | 91.85 | 177.91 | 0.139 | 2500 / 4050 |
| | BSPNet [8] | 18.38 | 5.32 | 93.13 | 112.65 | 0.077 | 587 / 1011 |
| | Polygen [37] | 47.86 | 4.18 | 81.42 | 52.48 | 0.025 | 185 / 558 |
| | MeshGPT | 53.88 | 3.94 | 65.73 | 19.91 | 0.004 | 150 / 288 |
| | Ours | **70.68** | **1.54** | **41.20** | **6.60** | **0.0005** | 216 / 418 |

meshes because it requires separate training of the vertex and face models. MeshGPT [45] may generate meshes with messy tessellation due to its limited capabilities and learning challenges. PivotMesh [59] can produce plausible results, but some artifacts may be observed in local details. Our method successfully generates shapes with complex topology and clean tessellation. The numerical results in Table 3 further validate the superiority of our approach in terms of generation diversity and fidelity, with most metrics surpassing those of other mesh generation methods. Table 2 presents metrics for different simplification ratios. We find that a $50\%$ simplification still yields decent results. The average generation time (full) is $\approx 1$ minute, suggesting a potential additional speedup of $50\%$.

We also compare our tokenization scheme with other tokenization methods, such as EdgeRunner [48] and BPT [60], in Table 4 under an unconditional setting. We observe that our method achieves comparable performance to the other methods when all ($100\%$) auto-regressive steps are applied. Notably, we find that it is considerably superior when only a few steps ($10\%$, $20\%$, $50\%$) are applied compared to the other methods. This is because the intermediate results produced by our scheme are consistently good approximations of the finest level of detail, whereas other approaches produce meshes in a partial-to-complete manner, causing their intermediate results to deviate significantly from the targets if the generation process is not fully completed.

In the last two columns of Table 4, we also present the average number of tokens required by each tokenization scheme, either per shape or per geometric element. We notice that our method, on average, needs fewer tokens to represent a mesh than other methods, even though our method generally requires more tokens to represent a vertex than other methods do to represent a triangle. This demonstrates the superiority of our tokenization scheme regarding compactness.

Table 4: Quantitative results of various tokenization approaches [48, 60] with different percentages of auto-regressive steps applied are measured on the bench category of the ShapeNet [2] dataset. EdgeRunner [48] and BPT [60] are based on triangles, while the elements in our method are vertices.

| Methods | COV ↑ | | | | MMD ↓ | | | | #Token ↓ | |
|---------|-------|-------|-------|-------|-------|------|------|------|-----------|-----------|
| | 10% | 20% | 50% | 100% | 10% | 20% | 50% | 100% | per shape | per elem. |
| EdgeRunner [48] | 9.43 | 30.99 | 40.33 | 54.90 | 10.32 | 4.87 | 2.59 | 0.81 | 5840 | 4.5 |
| BPT [60] | 15.15 | 26.65 | 34.11 | **56.35** | 3.02 | 1.97 | 1.26 | 0.76 | 3235 | **2.73** |
| Ours | **41.07** | **47.40** | **54.84** | 56.29 | **1.67** | **1.15** | **0.87** | **0.75** | **2556** | 9.1 |

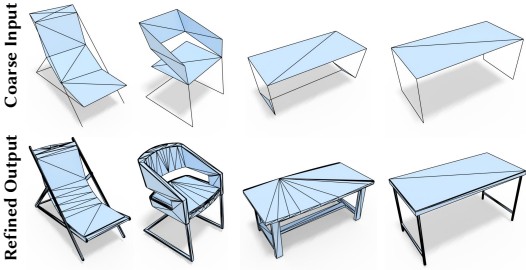

Figure 13: Our representation supports refining a coarse mesh input (top row) into a fine-grained output mesh (bottom row).

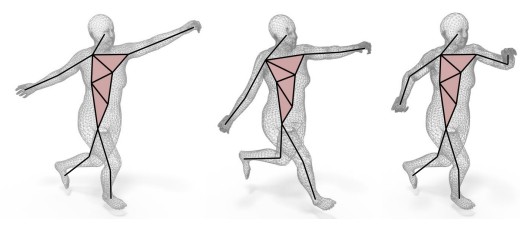

Figure 14: A coarse skeleton can be easily derived using our approach for fine-grained mesh editing using surface deformation.

## 5.4 Possible Applications with Our Approach

**Shape Refinement from Sketchy Inputs.** Our method can also be used to refine a 3D sketchy shape, as shown in Figure 13. This is practically useful for 3D modelers, as they only need to specify a coarse outline without expending effort on fine-grained geometric details, significantly saving time.

**Shape Editing by Manipulating a Coarse Skeleton.** A skeleton can be easily derived using our approach to facilitate the manipulation of a fine shape, as shown in Figure 14. By modifying the coarse skeleton, marked in red, one can easily manipulate the human shape through surface deformation.

## 6 Limitation, Future Work and Conclusion

Like prior works [45], the main drawback of our method is its relatively limited generalization ability compared to those built upon continuous space (*e.g.*, Diffusion). Using an order-of-magnitude larger amount of training data may alleviate this problem. To suppress excessive topological flexibility, we may use PSC to create an initial result with the desired topology and then use the normal progressive meshes representation [22] for the remaining steps of further refinement without changing the topology. During generation, we may reduce the linear complexity to logarithmic complexity by using parallel vertex splits [14, 23] in future work. Lastly, our method lacks gradients to update the PSC; therefore, it would be interesting to derive a differentiable PSC that can adaptively fit a mesh in a coarse-to-fine manner through local remeshing with gradient guidance.

This paper suggests formulating direct mesh generation in the order of levels of detail (LODs), consistent with how humans perceive objects, and generating in a preferable coarse-to-fine manner. The basic idea is to generalize meshes to simplicial complexes, reverse the mesh simplification process, and train a model to learn mesh generation, initially starting with a single point and iteratively refining the previous coarse results. Experiments verify the promise and effectiveness of our approach.

## Acknowledgment

We sincerely thank Prof. Zhen Liu for his valuable discussions regarding our work. This work was supported by the Key-Area Research and Development Program of Guangdong Province, China, under Grant 2024B0101040004 for the project titled "Research and Application of Common Key Technologies for Embodied Intelligence of Robots Based on AI Large Models."

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
