# OpenReview forum: "ARMesh: Autoregressive Mesh Generation via Next-Level-of-Detail Prediction"
_NeurIPS.cc/2025/Conference — NeurIPS 2025 poster_

### Official Review · Reviewer_L8JW · 2025-06-04

**Clarity:** 3
**Significance:** 3
**Originality:** 4
**Rating:** 5
**Confidence:** 5

**Summary:**

This paper proposes ARMesh, a novel autoregressive framework for direct 3D mesh generation that synthesizes mesh progressively from coarse to fine levels of detail (LOD), departing from conventional lexicographic face-by-face generation. Its core contributions are:
1. Insight:
Limitations in current AR mesh generators (e.g., MeshGPT, PolyGen) lies in constructing meshes in fixed orders unrelated to geometric hierarchy. ARMesh models the generation porcess by next level-of-detail(LOD) prediction compure to next scale generation in 2D image generation problem, which able to model the unorder 3D space.
2. Generalized Simplification (GSlim):
Introduces a topology-aware mesh simplification algorithm operating on simplicial complexes which generalizing meshes to include points/edges. It reduces any input mesh to a single point in n-1 steps, producing a fine-to-coarse LOD sequence.
3. Progressive Simplicial Complex (PSC):
Reverses GSlim into a generative coarse-to-fine process which enables the following AR mesh generation process.
4. Autoregressive Modeling:
  - Tokenizes PSC sequences into discrete vocabularies.
  - Trains a transformer decoder to predict next refinement tokens autoregressively.
  - Employs constrained decoding (DFS + hardcoded rules) ensuring topological validity.

**Questions:**

1. Could the authors evaluate ARMesh on a large-scale 3D dataset (e.g., Objaverse, ShapeNet-XL) to assess scalability and generalization? I fully understand that full training might be infeasible during rebuttal period, minimal experiments (e.g., fine-tuning on a subset) or qualitative results would suffice.
2. Could the authors compare ARMesh with SOTA autoregressive methods? I understand they are trained on some large-scale dataset like the objverse. Could author re-train them on identical dataset which paper use?
3. Could authors provide quantitative results on condition generation result? For example: coarse mesh/sparse point cloud conditioned.

**Ethical Concerns:**

["NO or VERY MINOR ethics concerns only"]

**Final Justification:**

The authors have solved my concerns, I will raise my score to 5.

**Limitations:**

1. Scalability to Real-World Complexity： The training and evaluation of ARMesh is limited to small-scale datasets. Its performance remains unverified on large-scale, diverse 3D collections (e.g., Objaverse. Without validation on high-complexity shapes or diverse topologies , claims of generalizability remain speculative. This restricts confidence in real-world applicability.
2. Unexplored Conditional Generation: While ARMesh intrinsically supports progressive refinement (Fig 13–14), its potential for conditional generation—e.g., text-guided LOD synthesis or point cloud completion—remains unexplored. The method’s coarse-to-fine paradigm is theoretically well-suited for such tasks (e.g., refining sketch inputs to detailed meshes).

**Quality:**

4

**Strengths And Weaknesses:**

Strengths
1. Originality & Significance:
  - Novel Next-LOD Prediction Paradigm: Proposes a fundamentally new autoregressive approach (Next-LOD prediction) for mesh generation. This departs significantly from prior lexicographic/traversal-order face-by-face generation, offering a more geometrically reasonable (coarse-to-fine) generation process. This is a conceptually original contribution to the field.
  - GSlim & PSC Framework: Introduces the generalized simplification algorithm (GSlim) enabling topology changes and reduction to a single point, and its reversible counterpart (PSC). This technical innovation is crucial for constructing the LOD sequences and is a significant advancement over traditional simplification (e.g., QSlim) for generative modeling.
2. Technically Sound Foundation:
  - The core methodology (GSlim simplification → PSC reversal → tokenization → AR transformer training) is well-motivated, rigorously described (esp. topological constraints and tokenization), and demonstrates clear technical competence.
3. Clarity:
  - Well-Structured and Readable: The paper is exceptionally well-written. The problem motivation (limitations of prior AR orderings), inspiration (2D next-scale prediction), proposed solution (LOD reversal), and technical details (GSlim, PSC, tokenization) are presented logically and clearly. Figures effectively illustrate concepts (e.g., vertex split, topology cases).


Weaknesses
1. Limited Scalability Demonstration:
Training and evaluation are conducted primarily on small-scale dataset. Crucially, recent state-of-the-art methods (e.g., EdgeRunner, MeshXL) leverage large-scale datasets like Objaverse. This omission raises concerns about the method’s scalability and generalization capability to complex, diverse shape distributions.
2. Incomplete State-of-the-Art Comparison:
While comparisons against PolyGen and MeshGPT are provided, Table 3 lacks critical benchmarks:
  - GET3D results for the Bench and Lamp categories
  - Contemporary baselines (e.g., EdgeRunner, MeshXL, MeshAnything) that report strong unconditional generation performance.
Validation against these methods on larger datasets (e.g., Objaverse) would significantly strengthen empirical claims.
3. Insufficient Ablation Analysis:
While ablation studies explore LOD ratios and penalty factors, key design choices lack rigorous validation:
  - Impact of transformer architecture/configurations (depth, width, attention mechanisms)
  - Tokenization alternatives beyond the proposed scheme
  - Quantitative and qualitative ablation results would better elucidate the contribution of individual components.

4. Narrow Scope in Conditional Generation:
The work focuses exclusively on unconditional generation. But it will be easy to adopt this method into condition mesh generation. Demonstrating efficacy in conditional settings (e.g., text-to-mesh, point cloud completion) would enhance the practical impact and versatility of the approach. Comparisons with conditional mesh generators are notably absent.

**If the authors solve my concerns during the rebuttal, I will increase my score to 5.**

---

> ### Author Rebuttal · Authors · 2025-07-31
>
> We are profoundly grateful to the reviewer for generously dedicating their time to provide us with such invaluable feedback. **In summary, we have decided to reserve additional experiments (e.g., conditional generation on large-scale datasets) for future work.** We truly appreciate your understanding of our intentions. Your understanding and support mean a great deal to us, and we would like to elaborate further on this.
>
> We wholeheartedly agree with the reviewer that providing additional experimental results on conditional generation (such as from images, texts, or point clouds) would significantly enhance the persuasiveness of our method’s effectiveness. However, we currently find ourselves constrained by limited computational resources and time, which regrettably prevents us from conducting these additional experiments that require large-scale training, as this process is indeed quite time-consuming and costly. We hope you can kindly understand our limitations.
>
> We also sincerely appreciate the reviewer’s thoughtful suggestion regarding minimal experiments (e.g., fine-tuning on a subset). However, we humbly feel that fine-tuning networks from other methods may present challenges for a fair comparison, as those networks have been trained on significantly more data than a network trained from scratch with a smaller dataset. We are grateful for your understanding of our perspective on this issue.
>
> In our current paper, we have conducted extensive ablation studies to explore the potential properties of our newly proposed representation, along with some unconditional generation results that serve as preliminary experiments demonstrating the efficacy of our progressive simplicial complex representation combined with deep learning for facilitating mesh generation in a coarse-to-fine manner. We humbly believe that scaling up to larger datasets and exploring diverse tasks can be effectively pursued in future studies. We sincerely appreciate your understanding, and we hope our work can still make a positive contribution to our community.
>
> Thank you once again, and we wish you a wonderful day!

---

> > ### Comment · Reviewer_L8JW · 2025-08-01
> >
> > Thank you for your response. I fully understand the constraints posed by limited computational resources, which make large-scale dataset experiments challenging. However, I still believe it is essential to conduct comparative experiments using datasets of the same scale. While the proposed next-LOD tokenization method is innovative, intuitively sound, and theoretically promising to outperform traditional mesh tokenization approaches like EdgeRunner, the authors should **AT LEAST** validate this claim through numerical experimental results. One feasible alternative is to retrain a mesh generation model using EdgeRunner-like tokenization, while maintaining the same model size and dataset scale as employed in the ARMesh paper—such an experiment would significantly strengthen the persuasiveness for both readers and reviewers.
> >
> > Additionally, as I previously noted, the authors should supplement ablation studies on the model design of ARMesh. Importantly, this type of ablation study generally not be constrained by computational resources.

---

> > > ### Author Response · Authors · 2025-08-06
> > >
> > > We are truly grateful for your kind understanding regarding the challenges we faced in conducting large-scale experiments. We will clarify and provide further experimental results as follows.
> > >
> > > **We would like to humbly summarize the main differences between our method and existing tokenization approaches:**
> > >
> > > 1. Our tokenization solution is intentionally designed to be simple and minimalist, simply discretizing the JSON information as illustrated in line 154. In contrast, other methods are specifically designed to be compact and often rely on the face traversal order.
> > >
> > > 2. The representation we tokenize is fundamentally different: we concentrate on tokenizing PSCs, while others focus on tokenizing meshes. This distinction highlights that our primary contribution is advocating for the use of PSCs for progressive generation. In contrast, the main contributions of other papers focus on their tokenization methods, which may facilitate scaling up to meshes with a high face count. Therefore, we humbly acknowledge that our initial objective in writing this paper is not to "outperform traditional mesh tokenization approaches," but rather to illustrate the process of generating a mesh in a coarse-to-fine manner. We are grateful for the opportunity to contribute to this new area.
> > >
> > >
> > > ---
> > >
> > > Nevertheless, we are sincerely thankful for the opportunity to share some results that compare our approach with other tokenization methods. Before we proceed, **we would like to provide brief overviews of these methods**.
> > >
> > > 1. **EdgeRunner**'s primary contribution lies in designing a tokenization algorithm that maximizes edge sharing between adjacent triangles. The fundamental idea is to traverse faces using the half-edge data structure, which we note is done in a face-by-face generation manner. Additionally, a minor contribution of their paper is the use of an extra "face count" token for complexity conditioning. Given that they categorize complexity into only three levels (low, medium, and high), they achieve three distinct levels of complexity in their generation results. Although they incorporate complexity conditioning, they fundamentally treat different complexities of the same shape as distinct meshes. This is evident in their tokenizations, which vary significantly from one another, even when approximating the same shape.
> > >
> > > 2. **BPT** 's main contribution is also a tokenization algorithm that employs block indices to compress coordinates and utilizes star-shaped patches for face compression. They generate meshes in a face-by-face manner, and we note that they only support one level of generated complexity.
> > >
> > > ---
> > >
> > > **We are pleased to share some of the experimental setups here**. To ensure fairness, we utilize the same network architecture, a 12-layer standard transformer network with a width of 768, for training. Each method's network is trained on 4 H20 GPUs for approximately 4 days. The training is conducted on the bench category of the ShapeNet dataset under an unconditional setting. For fairness, we did not train EdgeRunner using decimated meshes; instead, we used the original input meshes. For additional details, we adhere to the same configurations as outlined in our submitted paper.
> > >
> > > **We present the quantitative experimental results in the tables below**, reporting the Coverage (COV) and Minimum Matching Distance (MMD), respectively. The horizontal percentages (10%~100%) indicate the proportion of auto-regressive steps applied; thus, 100% corresponds to the final generation results. As shown in the results below, as the meshes are generated, they become closer to the target shape distribution, with COV gradually increasing and MMD gradually decreasing as more tokens are generated during inference. However, we observe significantly steadier changes in our method, where COV maintains noticeably higher values (consistently > 40) and MMD remains at relatively lower values (consistently < 2). Comparative methods, such as EdgeRunner and BPT, exhibit larger fluctuations, indicating that their intermediate generation results are far from ideal. Our method also achieves comparable results with EdgeRunner and BPT when 100% generation steps are applied. This experiment reveals that our tokenization solution is not only among the state-of-the-art methods but also capable of delivering progressive coarse-to-fine results that consistently mimic the target shape during auto-regressive generation.
> > >
> > > **COV**: (the higher the better)
> > > | Methods | 10% | 20% | 50% | 100% |
> > > |-----|-----|-----|-----|-----|
> > > | EdgeRunner | 9.43 | 30.99 | 40.33 | 54.90 |
> > > | BPT              | 15.15 | 26.65 | 34.11 | **56.35** |
> > > | Ours             | **41.07** | **47.40** | **54.84** | 56.29 |
> > >
> > >
> > > **MMD**: (the lower the better)
> > > | Methods | 10% | 20% | 50% | 100% |
> > > |-----|-----|-----|-----|-----|
> > > | EdgeRunner | 10.32 | 4.87 | 2.59 | 0.81 |
> > > | BPT | 3.02 | 1.97 | 1.26 | 0.76 |
> > > | Ours | **1.67** | **1.15** | **0.87** | **0.75** |

---

> > > ### Author Response · Authors · 2025-08-06
> > >
> > > **(continued)**
> > >
> > > ---
> > >
> > >
> > > To investigate the **compression ratio** achieved by different tokenization methods, we humbly tokenized objects from the ShapeNet dataset and present the average number of tokens required in the table below. The first table showcases the average number of tokens needed for shape tokenization, while the subsequent table presents the average number of tokens required to encode each geometric element (either face or vertex). We would like to emphasize that, since our method is vertex-based, we report the per-vertex metric in the second table. We are grateful to observe that our tokenization method yields significantly shorter sequences, even though the average tokens per vertex are higher than those of face-based tokenization methods.
> > >
> > >
> > > **#AvgTokenPerShape**: (the lower the better)
> > > | EdgeRunner  | BPT  | Ours  |
> > > |-----|-----|-----|
> > > | 5840 | 3235 | **2556** |
> > >
> > > **#AvgTokenPerElem**: (the lower the better)
> > > | EdgeRunner (per face)  | BPT (per face)  | Ours (per **vertex**)  |
> > > |-----|-----|-----|
> > > | 4.5 | **2.73** | 9.1 |
> > >
> > >
> > > In this paragraph, we would like to share some experimental details regarding our study. We adhere to the standard techniques in LLM tokenization, employing Byte Pair Encoding (BPE) to combine tokens. Our vocabulary size reaches 16,384 following the application of BPE. The application of BPE results in an impressive compression ratio of approximately 2.6 times. Furthermore, we wish to highlight that our coordinate resolution is significantly higher than that of other methods, achieving a spatial resolution of 65,536, made possible by the high-precision representation using fp16 byte encoding. In contrast, other approaches, such as EdgeRunner, typically achieve a spatial resolution of only 512.
> > >
> > >
> > >
> > > ---
> > >
> > >
> > >
> > > While we are grateful for your suggestions, **we humbly find ourselves uncertain about how your proposed ablation studies might contribute to and strengthen the claims made in our paper**.
> > > 1. You suggested examining the impact of transformer architecture and configurations (such as depth, width, and attention mechanisms). However, in our experiments, we opted for the most standard transformer architecture. We respectfully believe that exploring these network choices diverges significantly from our initial focus, as our design is entirely independent of the specific network utilized, provided that it is an auto-regressive model. We have chosen to use the standard transformer architecture due to its recent popularity and widespread adoption in the field. Additionally, we would like to clarify that we have not conducted any theoretical analysis regarding network size or scaling laws, as these topics are not central to our paper. Consequently, we do not see a necessity for experiments related to depth and width, which fall outside the primary discussion points of our work. We sincerely appreciate your understanding in this matter.
> > > 2. Regarding the ablation studies on tokenization alternatives, we humbly acknowledge that, as the first to propose tokenizing PSCs (not meshes) within deep learning, there are currently no existing PSC tokenization alternatives in the field. Nevertheless, we respectfully compare our vertex-based tokenization method with several existing face-based tokenization solutions, as detailed in the earlier paragraphs. We believe that more effective PSC tokenization alternatives may exist and have yet to be explored, and we recognize that this is an entirely different research topic that deserves further investigation in the future. We are sincerely grateful for your understanding of this matter.
> > >
> > > **We would like to humbly remind you that we have conducted extensive ablation studies on our primary focus, the PSC representations (see Section 5.1), along with their learning properties and dynamics (Sections 5.2 and 5.3) in our submitted version.** We presented these results, and we sincerely believe they offer valuable support for the claims made in our paper.
> > >
> > >
> > > At the end, we sincerely appreciate your understanding and support.

---

> > > > ### Comment · Reviewer_L8JW · 2025-08-08
> > > >
> > > > I thhink this rebuttal totally solved my concerns, I will rasie my score to 5. Thanks for author's reply!

---

### Official Review · Reviewer_1KDk · 2025-07-01

**Clarity:** 4
**Significance:** 2
**Originality:** 3
**Rating:** 4
**Confidence:** 5

**Summary:**

The paper proposes a novel method to generate meshes in a coarse-to-fine manner. To construct a compatible representation for arbitrary topology, especially the starting point, the authors generalize 3D meshes to simplicial complexes and adapt the commonly used mesh simplification algorithm to GSlim. By tokenizing the vertex split operation, the paper parses each mesh as a 1D refining sequence that can be further modeled with a transformer-based AR model.

**Questions:**

I am generally positive about the paper. The paper proposes a good way to model meshes in a coarse-to-fine way. But the tokenization process seems too complicated, making the sequence quite long. I think it's a big limitation on how far we can go. What do we get out of making such sacrifices, in other words, are there some potential solutions, or how can we utilize the intermediate results? What is the difference between generating the whole mesh and them simplifying it, and the proposed method?

Other comments:
- The title of section 3.1: simplical -> simplicial

**Ethical Concerns:**

["NO or VERY MINOR ethics concerns only"]

**Final Justification:**

The rebuttal addresses most of my concerns except the comparison against strong baselines (edgerunner, BPT). I will keep my original rating.

**Limitations:**

Yes.

**Paper Formatting Concerns:**

No.

**Quality:**

3

**Strengths And Weaknesses:**

Strengths:
- The method to tokenize the simplification operation is non-trivial and well-designed.
- The overall writing is good, making the technical details clear.

Weaknesses:
- For a mesh with N vertices, the length of mesh sequences is roughly larger than $10N$ or even longer, which is longer than per-face per-coordinate tokenization. This may hinder the model ability to generate complex meshes.
- Compared to direct mesh generation, it is not entirely clear how we can use the intermediate results for some potential applications. Although there are some mentions, no detailed algorithms or results are given in the article.
- The dataset and evaluation are limited. At least, edgerunner, meshanything, BPT are quite relevant but they are not shown in the quantitative results. Shapenet is not diverse enough for mesh generation.

---

> ### Author Rebuttal · Authors · 2025-07-31
>
> We are sincerely grateful to the reviewer for generously dedicating their time to provide us with such constructive feedback. We are committed to addressing your questions in a manner that we hope will effectively resolve your concerns. In line with this year's rebuttal policy, we will provide a textual response only, without any external links. Thank you for your understanding and support.
>
> 1. **The tokenization process seems too complicated, making the sequence quite long.** Thank you for highlighting this limitation! We would like to clarify that our paper does not specifically focus on proposing an effective tokenization method; rather, we present a straightforward solution to make it work (as noted in footnote 4 at the bottom of page 6). In practice, we adapted techniques from LLMs to reduce the number of tokens, employing Byte Pair Encoding (BPE) for compression in our experiments. On average, this approach can reduce token lengths by approximately 2$\sim$3 times.
> For example, a mesh with 500 vertices and 1,000 faces (where $F \approx 2V$) and an average tokenized length of 10 would require $ (500-1) \times 10 = 4,990 $ tokens to represent a highly complex mesh. However, with BPE, this number can be significantly reduced to about 2k tokens. In comparison, a baseline method using per-face per-coordinate encoding would need $ 1,000 \times 9 = 9,000 $ tokens, which can also be reduced to around 2k tokens using BPT [1].
> It is evident that the compression ratio of our method is comparable to recent strong compression baselines, which underscores a major benefit of using progressive representation for compression, as highlighted in [2]. We truly appreciate your understanding and support in this matter.
>
>
> 2. **How can we utilize the intermediate results?** We appreciate your feedback on this aspect! One of the key advantages of our representation is the ability to obtain intermediate (coarse) results without needing to generate the mesh with the highest complexity. For instance, consider a simple case where our ground truth is a mesh with 500 vertices and 1,000 faces. Conventional methods may require a total of 1,000 steps for generation; if fewer than 1,000 steps are taken, the resulting mesh may appear incomplete, with some faces missing, making it less representative of the ground truth.
> In contrast, our method allows us to obtain the ground truth mesh in just 499 steps. Moreover, in many cases, we do not require a mesh of such high complexity. For example, if we only need a simplified version, we can perform even fewer steps (e.g., 100 steps), and the resultant mesh will still closely resemble the original ground truth. By reformulating mesh generation in a coarse-to-fine manner, we can effectively control both geometric complexity and the time and computational budget required to generate a mesh with the desired complexity.
> Finally, the coarse mesh can be utilized in various scenarios, such as game development and streaming or previewing the generation process. We sincerely appreciate your concerns and support in this discussion.
>
>
> 3. **What is the difference between generating the whole mesh and them simplifying it, and the proposed method?** We sincerely appreciate your attention to this point! It is indeed a fascinating aspect. The primary advantage of our approach is its ability to save both time and computational resources. When the network is trained with a high-fidelity mesh dataset, conventional methods often require an excessive number of generation steps to produce high-fidelity meshes, making the process quite costly. Additionally, an extra simplification process is needed if a low-poly result is desired.
> While one could train multiple models with varying mesh complexities to partially address this limitation, we believe this is not the ultimate solution. Our approach allows for greater control over the time and computational budget required for generation, enabling us to achieve results with fewer inference steps. We hope our method can make a positive contribution to our community, and we are truly grateful for your kind support and understanding.
>
> ---
>
> [1] Haohan Weng, et al. Scaling Mesh Generation via Compressive Tokenization. CVPR 2025.
>
> [2] Hugues Hoppe. Progressive meshes. SIGGRAPH 1996.

---

> > ### Comment · Reviewer_1KDk · 2025-08-02
> >
> > I appreciate the response from the authors. My concerns about the sequence length and the potential application are almost addressed.  But I think the comparisons against strong baselines (edgerunner, BPT) are essential, at least on shapenet, just as other reviewers indicated, which does not require large computational resources. And it will be nice to include how the byte pair encoding is used to reduce the sequence length in the final version. I did not see any mention in the submission.

---

> > > ### Author Response · Authors · 2025-08-06
> > >
> > > We are genuinely grateful for your support of our work! We conducted comparative experiments with recent strong baselines, such as EdgeRunner and BPT, using the same setup.
> > >
> > > **We present the quantitative experimental results in the tables below**, reporting the Coverage (COV) and Minimum Matching Distance (MMD), respectively. The horizontal percentages (10%~100%) indicate the proportion of auto-regressive steps applied; thus, 100% corresponds to the final generation results. As shown in the results below, as the meshes are generated, they become closer to the target shape distribution, with COV gradually increasing and MMD gradually decreasing as more tokens are generated during inference. However, we observe significantly steadier changes in our method, where COV maintains noticeably higher values (consistently > 40) and MMD remains at relatively lower values (consistently < 2). Comparative methods, such as EdgeRunner and BPT, exhibit larger fluctuations, indicating that their intermediate generation results are far from ideal. Our method also achieves comparable results with EdgeRunner and BPT when 100% generation steps are applied. This experiment reveals that our tokenization solution is not only among the state-of-the-art methods but also capable of delivering progressive coarse-to-fine results that consistently mimic the target shape during auto-regressive generation.
> > >
> > > **COV**: (the higher the better)
> > > | Methods | 10% | 20% | 50% | 100% |
> > > |-----|-----|-----|-----|-----|
> > > | EdgeRunner | 9.43 | 30.99 | 40.33 | 54.90 |
> > > | BPT              | 15.15 | 26.65 | 34.11 | **56.35** |
> > > | Ours             | **41.07** | **47.40** | **54.84** | 56.29 |
> > >
> > >
> > > **MMD**: (the lower the better)
> > > | Methods | 10% | 20% | 50% | 100% |
> > > |-----|-----|-----|-----|-----|
> > > | EdgeRunner | 10.32 | 4.87 | 2.59 | 0.81 |
> > > | BPT | 3.02 | 1.97 | 1.26 | 0.76 |
> > > | Ours | **1.67** | **1.15** | **0.87** | **0.75** |
> > >
> > >
> > > ---
> > >
> > > We also conducted experiments to compare our vertex-based tokenization method with face-based tokenization approaches, such as EdgeRunner and BPT, and we present the results below. **We will definitely include a detailed explanation of the commonly used BPE technique in our revised version.**
> > >
> > > The first table showcases the average number of tokens needed for shape tokenization, while the subsequent table presents the average number of tokens required to encode each geometric element (either face or vertex). We would like to emphasize that, since our method is vertex-based, we report the per-vertex metric in the second table. We are grateful to observe that our tokenization method yields significantly shorter sequences, even though the average tokens per vertex are higher than those of face-based tokenization methods.
> > >
> > >
> > > **#AvgTokenPerShape**: (the lower the better)
> > > | EdgeRunner  | BPT  | Ours  |
> > > |-----|-----|-----|
> > > | 5840 | 3235 | **2556** |
> > >
> > > **#AvgTokenPerElem**: (the lower the better)
> > > | EdgeRunner (per face)  | BPT (per face)  | Ours (per **vertex**)  |
> > > |-----|-----|-----|
> > > | 4.5 | **2.73** | 9.1 |
> > >
> > > ---
> > >
> > > **For more experimental details regarding these experiments, please refer to our discussion with Reviewer L8JW.**

---

### Official Review · Reviewer_KceG · 2025-07-01

**Clarity:** 3
**Significance:** 3
**Originality:** 3
**Rating:** 4
**Confidence:** 4

**Summary:**

This paper proposes a novel autoregressive (AR) framework for 3D mesh generation that constructs meshes in a coarse-to-fine manner, departing from the conventional face-by-face generation paradigm. Inspired by next-scale prediction in 2D image generation, the authors model the mesh generation process as the reverse of a mesh simplification procedure. They leverage a transformer-based AR model to reconstruct the mesh, starting from a single vertex and gradually increasing geometric detail without a fixed mesh topology. The method improves mesh quality and supports downstream tasks such as mesh refinement and editing. The method is novel in the field of mesh generation as it's the first next-scale prediction mesh generation method.

**Questions:**

1. **Training and inference speed.**
   How does the training and inference speed of the proposed method compare with face-by-face generation methods such as MeshGPT?

2. **Extension to conditional generation.**
   Can the proposed framework support conditional refinement or editing tasks, such as conditioning on an image or partial point cloud?

3. **Failure rate.**
   What is the failure rate of the method during generation? Prior face-by-face methods often exhibit limited success when generating high-face-count meshes—how does this method compare?

4.  Lack of comparison experiments with recent strong baselines and the absence of experiments on conditional generation.

**Ethical Concerns:**

["NO or VERY MINOR ethics concerns only"]

**Final Justification:**

I maintain my positive rating.

**Limitations:**

yes

**Paper Formatting Concerns:**

nil

**Quality:**

3

**Strengths And Weaknesses:**

## Strengths

1. **Novelty.**
   The paper introduces a new perspective on mesh generation by reversing the mesh simplification process and formulating it as a progressive coarse-to-fine autoregressive task. To the best of our knowledge, this is the first work to adopt a next-scale prediction approach for meshes, analogous to recent advances in 2D generation.

2. **Application.**
   The proposed coarse-to-fine generation process naturally enables applications such as mesh refinement and editing, which are challenging to achieve with conventional face-by-face generation schemes.

3. **Clarity.**
   The paper is well-written and provides a thorough description of the proposed method.

## Weaknesses

1. **Limited comparison.**
   The paper lacks comprehensive comparison with other recent mesh generation methods such as MeshXL, MeshAnything, and DeepMesh, all of which are publicly available.

2. **Lack of conditional generation.**
   The experiments are limited to unconditional generation. Given that applications like mesh refinement and editing naturally arise in conditional settings, it is important to validate the proposed method under such scenarios (e.g., conditioned on an image or point cloud).

---

> ### Author Rebuttal · Authors · 2025-07-31
>
> We are sincerely grateful to the reviewer for generously dedicating their time to provide us with such constructive feedback. We are committed to addressing your questions in a manner that we hope will effectively resolve your concerns. In line with this year's rebuttal policy, we will provide a textual response only, without any external links. Thank you for your understanding and support.
>
> 1. **Training and inference speed.** This aspect is indeed quite practical, and we appreciate the opportunity to discuss it. Both our method and MeshGPT utilize auto-regressive techniques, but the primary distinction lies in what determines the generation complexity: our approach is based on the number of vertices $V$, while MeshGPT depends on the number of faces $F$. Typically, for a mesh, there is an approximate relationship of $F \approx 2V$, which suggests that our method may require fewer geometric elements for generation. In our experiments, when generating 100 meshes randomly and unconditionally, *our method generally takes about 60 seconds per shape, whereas MeshGPT averages around 80 seconds*. We would like to humbly emphasize that these speed metrics were obtained using a single H20 GPU. Regarding training speed, it largely depends on the computational resources available; generally, longer training times lead to better generalization. For our model, we typically require about 16 GPU-days, while MeshGPT necessitates a total of 24 GPU-days to complete its training. Lastly, we wish to acknowledge that actual performance may vary slightly across different platforms and machines. We appreciate your understanding in this matter.
>
> 2. **Extension to conditional generation.** Thank you for highlighting the importance of conditional generation tasks! They are indeed invaluable for practical scenarios. We believe that our proposed framework has the potential to support additional tasks, such as conditional refinement, editing, or generation from images or point cloud inputs. To accommodate different modalities, we may consider established conventions that convert images or texts into tokens using ViT or BERT for feature extraction, subsequently fusing these features into our transformer via cross-attention mechanisms.
> We must humbly acknowledge that achieving these tasks requires very large models and substantial amounts of data. Unfortunately, we are currently limited by our computational resources and time, making it challenging to train large-scale models on extensive datasets during the rebuttal period. We sincerely apologize for this limitation and remain committed to pursuing this direction by extending our methods to perform conditional generation tasks in the future. We sincerely appreciate your understanding and consideration. Thank you for your support.
>
> 3. **Failure rate.** Thank you for highlighting the aspect we previously overlooked. Indeed, there are notable works in the related field aimed at supporting the generation of meshes with a high face count, such as Meshtron, which claims the ability to generate meshes with up to 10k faces, significantly more than what our current study addresses.
> In our experiments, we found that the failure rate of our method is acceptable, remaining below 5% across 100 randomly generated objects. This metric was measured on the ShapeNet dataset, which includes shapes with fewer than 1,700 vertices and 800 faces each. We sincerely acknowledge that we currently lack the computational resources and time necessary to conduct large-scale experiments focused on generating high face counts. As such, we have positioned our paper to explore the potential advantages of using our progressive simplicial complex representation before scaling up to larger models or datasets. We appreciate your understanding in this matter.
> Additionally, we have observed that using our trained model for extrapolation, specifically, performing additional refinement steps with training on a low face-count dataset, is not feasible. This limitation arises because our tokenization process includes the vertex index, which restricts our model's capabilities based on the vertex count. As a result, the model has not encountered cases with a high number of vertices, leading to poor generalization when applied to meshes with high face counts. We are grateful for your understanding of this limitation. We will certainly discuss this point in the limitation section of our revised paper.
> Finally, we would like to share a strategy for managing failure cases. We have observed that unsatisfactory results often become apparent within the first few generation steps. For instance, after generating just 10 steps, we can often determine whether the current generation will yield acceptable results. If the produced outputs show low confidence in generation (as indicated by the classification probability of the generated tokens), we can recognize a potential failure early and stop the generation process. Thank you once again for your valuable feedback.
>
> 4. **Lack of comparison experiments with recent strong baselines and the absence of experiments on conditional generation.** We sincerely apologize for not including comparisons with recent strong baselines (such as publicly available methods like MeshXL, MeshAnything, and DeepMesh) on the tasks of conditional generation from texts, images, or point cloud inputs. We are committed to discussing and citing these methods, as well as presenting our limitations, in our revised paper. We must acknowledge that, due to limited computational resources and time, we are currently unable to perform these tasks, as they require training on a very large dataset and utilizing a large-scale model. Instead, we chose to focus on unconditional generation tasks, which our baseline method, MeshGPT, also addresses. Comparing our results with MeshGPT is more feasible for our experiments, given the lower resource requirements. Additionally, we positioned our paper to explore the benefits of using the newly developed progressively simplicial complex representation before scaling up to such large datasets and models. We genuinely hope for your understanding regarding this limitation and greatly appreciate your support.

---

> > ### Comment · Reviewer_KceG · 2025-08-04
> >
> > I appreciate the authors’ response. My questions have been well addressed. However, I remain concerned about the comparisons with stronger baselines, as raised by other reviewers. I will keep my original rating.

---

> > > ### Author Response · Authors · 2025-08-06
> > >
> > > We are genuinely grateful for your support of our work! We conducted comparative experiments with recent strong baselines, such as EdgeRunner and BPT, using the same setup.
> > >
> > >
> > > **We present the quantitative experimental results in the tables below**, reporting the Coverage (COV) and Minimum Matching Distance (MMD), respectively. The horizontal percentages (10%~100%) indicate the proportion of auto-regressive steps applied; thus, 100% corresponds to the final generation results. As shown in the results below, as the meshes are generated, they become closer to the target shape distribution, with COV gradually increasing and MMD gradually decreasing as more tokens are generated during inference. However, we observe significantly steadier changes in our method, where COV maintains noticeably higher values (consistently > 40) and MMD remains at relatively lower values (consistently < 2). Comparative methods, such as EdgeRunner and BPT, exhibit larger fluctuations, indicating that their intermediate generation results are far from ideal. Our method also achieves comparable results with EdgeRunner and BPT when 100% generation steps are applied. This experiment reveals that our tokenization solution is not only among the state-of-the-art methods but also capable of delivering progressive coarse-to-fine results that consistently mimic the target shape during auto-regressive generation.
> > >
> > > **COV**: (the higher the better)
> > > | Methods | 10% | 20% | 50% | 100% |
> > > |-----|-----|-----|-----|-----|
> > > | EdgeRunner | 9.43 | 30.99 | 40.33 | 54.90 |
> > > | BPT              | 15.15 | 26.65 | 34.11 | **56.35** |
> > > | Ours             | **41.07** | **47.40** | **54.84** | 56.29 |
> > >
> > >
> > > **MMD**: (the lower the better)
> > > | Methods | 10% | 20% | 50% | 100% |
> > > |-----|-----|-----|-----|-----|
> > > | EdgeRunner | 10.32 | 4.87 | 2.59 | 0.81 |
> > > | BPT | 3.02 | 1.97 | 1.26 | 0.76 |
> > > | Ours | **1.67** | **1.15** | **0.87** | **0.75** |
> > >
> > >
> > > **For more experimental details regarding these experiments, please refer to our discussion with Reviewer L8JW.**

---

### Official Review · Reviewer_HBFZ · 2025-07-03

**Clarity:** 2
**Significance:** 3
**Originality:** 3
**Rating:** 5
**Confidence:** 3

**Summary:**

This paper formulates the triangle mesh generation process as an autoregressive (AR) prediction on the next level-of-detail. The key idea is to ask the AR model to learn the mesh “densification” process (reverse of simplification) while the simplified results are provided by mesh simplification method geometry processing. The produced sequence naturally forms a sequence of increasing levels of detail. The challenge is to “translate” the full sequence of mesh densification to tokens that are compatible with the AR model. For that purpose, the paper proposes a generalized mesh simplification method and designs a method to serialize the vertex split operation as a token. With the train AR model, the resulting system is able to produce a realistic sequence of generation. The authors also show results of mesh simplification (with the proposed modified simplification method), mesh generation, fitting and application such as editing and guided generation given user input.

**Questions:**

1. “Virtual Edges” in (a bit after than) Line 136: I like the idea to include topological change.
- (a) How are these virtual edges represented in the later tokenization process? The paragraph in Line 159-171 is great, walking the audience through the newly defined concepts. It would be great to describe other examples (in particular, corner case scenario) with some examples (maybe in the supp. mat.).
- (b) While flexible, allowing topology changes can also open a box of worms, enabling the model to create a mesh with unacceptable topologies. Will the proposed system suffer from this? How to prevent it?


2. I appreciate Section 4.2 as it tries to define the vertex split operation and formally “serialize” the operation as a symbol to be compatible with later AR model learning processes. The paper also explores properties of the topological constraints. Are the four rules in Line 180 complete? How to show if the rules here are complete.


3. The example in Line 154 is not fully explored. The “topo” list: I see that those integers refer to the case ID in Table 1 (and Line 166). So those values in that example are not corresponding to the example in Fig 4? This part is a bit confusing. Currently I assume the content in Line 154 is just arbitrary and not connected to later paragraphs.


4. It’s a missed opportunity to use the newly defined rules/topo case in Line 167-184 to explain the example in Fig. 4. The “Topological List” in Fig 4 is still not easy to understand.


5. The fitting experiment in Fig 11: can the authors provide a more detailed description of how this experiment is done? What loss function/process to guide the generation towards the target bunny mesh? What starting point (single vertex, or a rough shape of the bunny mesh) for the generation?


6. Any failure case examples? The limitation section in Line 292 mentioned the limited generalization issue. Can we see some examples? Will it have geometry artifacts and/or topology issues?

**Ethical Concerns:**

["NO or VERY MINOR ethics concerns only"]

**Final Justification:**

The rebuttal addressed my concerns. The method presents novel ideas, solid reasoning process and good results. I will improve my rating to "acceptance".

**Limitations:**

I’m concerned about the robustness of the method. Can it always produce nice geometry and topology? See my comment in Question section item 1 and 6.


The authors mentioned the generalization issue in the limitation section. Can we see some examples and do some analysis?

**Paper Formatting Concerns:**

Fig 11 and Tab 2 seem out of the boundary of the PDF file.


Maybe the title of section 5.1 shouldn’t be “ablation studies”, but “Generalized Simplification GSlim” or so.

**Quality:**

3

**Strengths And Weaknesses:**

Strength:
- Very interesting formulation to convert the progressive generation process of triangle mesh into autoregressive sequence prediction.
- The key element of the paper is how to convert progressive mesh generation (vertex split operation) into an AR model compatible token.


Weakness:
- I’m concerned about how robust the generated results are. In particular, will the mesh have topology issues since the model allows flexible change of topology.
- The authors put quite some effort on explaining the tokenization process. But there are still quite some parts that are not very clear. This would be key to this paper, and it would be great to address my questions, as those might be the questions for future audiences.

---

> ### Author Rebuttal · Authors · 2025-07-31
>
> We are truly grateful to the reviewer for dedicating their valuable time to provide us with such constructive feedback. We sincerely appreciate your insights and will now address your questions, hoping that our responses will adequately address your concerns. In accordance with this year's rebuttal policy, we will provide a textual response only, without any external links. Thank you once again for your understanding and support.
>
> 1. We are truly thankful to the reviewer for their kind appreciation. We wholeheartedly agree that introducing topological changes allows any chosen mesh to be simplified to a single point, serving as a universal starting point for generation. Your recognition of this point is deeply appreciated and means a great deal to us.
>
>    **(a) How are these virtual edges represented in the later tokenization process?** The presence of a (virtual) edge is encoded in the topological list, with the first element indicating whether the edge exists. Specifically, the case V0 in Table 1 signifies that the edge is virtual, while V1 indicates that the edge is real and will be present after the vertex split. The topological list is then tokenized into a sequence of classification tokens, as illustrated in Figure 5. We sincerely recognize that our explanation of these new concepts may not be fully sufficient, and we will provide additional clarifications and examples in the supplementary materials to enhance understanding. We truly appreciate your understanding and support.
>
>    **(b) Flexible topology changes.** Thank you for your insightful observation! We are grateful for the insights that guide us in this direction. At each vertex split step, we have the flexibility to modify the topology of neighboring simplices as needed. For instance, in the case of a mesh with 100 vertices, where each vertex has an average of 10 neighboring simplices, we estimate an upper bound of $10 \times (100 - 1) = 990$ opportunities to modify the local topologies. It’s important to note that this is an upper bound, as the topological constraints outlined by the four rules will slightly reduce this number. This observation underscores the remarkable flexibility of our approach. However, we have observed some unacceptable changes when utilizing a poorly trained model with limited generalization ability. This has led to generated meshes that appear random and do not reflect typical shapes. To address this challenge, we believe that a straightforward yet effective solution is to ensure that the model is well-trained. Additionally, we have some ideas that may address this problem more comprehensively. For instance, we could allow the model to predict a coarse mesh with 10 vertex splits that introduce all the essential topological changes contributing to the overall topology, while the remaining 90 vertex splits would maintain the overall topology. In this scenario, our model would effectively reduce to a "progressive mesh" method (rather than a "progressive simplicial complex" method), introducing mesh subdivision without topological changes, thus enabling us to better control the complexity of these changes. We view this as an important direction for future work, and we appreciate your guidance in this matter.
>
> 2. **Are the four rules in Line 180 complete?** We are truly grateful for your interest in these four rules! We humbly believe that these four rules are complete, serving as both sufficient and necessary conditions. We have carefully derived them by thoroughly enumerating all possible cases in tables and summarizing the rules that encompass each entry. For those who are interested, we would be glad to provide a rigorous proof in the supplementary material. In a broader sense, our proof can be viewed as somewhat analogous to deriving logical expressions from truth tables.
>
> 3. **So those values in that example are not corresponding to the example in Fig 4?** The topological list $[1, 2, 5, 3, 3, 4, 6, 7]$ presented in line 154 corresponds to Figure 4. As indicated in lines 165~166, this list can be converted to: V1, E0, E3, E1, E1, E2, F0, F1, which describe the topological changes of the eight simplices in the left subfigure of Figure 4. For instance, vertex 0 is classified as V1 because a new edge 5 is created following the vertex split; face 1 is categorized as F1 because it was originally connected to vertex 0, but after the split, it connects to the new vertex. We also observe that the lengths of these two lists are equal, reflecting the number of star simplices, which includes vertex 0 itself along with its adjacent edges and faces. We appreciate your attention to these details!
>
> 4. **It's a missed opportunity to use the newly defined rules/topo case in Line 167-184 to explain the example in Fig. 4.** We have provided explanations for the topological cases in Figure 4, along with examples in lines 172$\sim$174, and we briefly discuss the reasons for the existence of these rules in lines 174$\sim$178. However, we acknowledge that the few examples presented in the current paper may not be sufficient, and we intend to provide further clarifications. For the zero-dimensional simplex, we focus on the selected vertex 0. We observe that after the split, it creates an edge 01 that connects the original vertex 0 to the newly created vertex 1, which corresponds to the topological case V1 in Table 1. Additionally, for edge 3, which originally connects to vertex 0, after the split, it is no longer connected to vertex 0 but instead to vertex 1. This indicates a transition from the source vertex to the target vertex, representing the topological case E1. Similarly, face 1 falls under the topological case F1, as it is now incident to the new vertex 1, as shown in Table 1. Regarding the constraint rules discussed in lines 180~184, we will provide an example and focus on the second rule for further explanation. If there are two edges, e1 and e2, incident to the split vertex and they are subsets of a face with a topological label F0, then the topological labels of e1 and e2 cannot be E1. This is because a face labeled F0 implies that its sub-simplices (e1 and e2) must also be incident to the source vertex \(s\), which contradicts the case E1 where the edge is connected to the target vertex \(t\). We appreciate your patience and understanding, and we will elaborate further in the supplementary materials, including an additional video to illustrate this point more clearly. Thank you for your interest!
>
> 5. **The fitting experiment in Fig 11: can the authors provide a more detailed description of how this experiment is done?** Let us provide some additional details regarding the experiments presented in Figure 11. DMesh is a differentiable Delaunay triangulation method that learns from a mesh input. We use the target mesh as input and allow DMesh to perform optimization, with its intermediate outputs reflecting its progressive learning outcomes.
> EdgeRunner, on the other hand, is a large-scale pretrained direct mesh generation network that takes a point cloud as input and produces a mesh in a predefined scanning order (not in a coarse-to-fine manner during the generation process). However, it offers only three levels of complexity conditioning, 1k, 2k, and 4k, without options for other complexities, such as 3k. Consequently, we utilize their method by feeding a point cloud as geometry conditioning and the complexity conditioning token (1k, 2k, or 4k) as input, resulting in a mesh output.
> We would like to note that although it is claimed that a 4k conditioning should ideally produce a mesh with 4k faces, our observations indicate that the generated mesh contains approximately only 2k faces. This suggests that their method may not effectively control geometric complexity, even when provided with an additional face count (complexity) token as input.
> Finally, our approach is similar to that of DMesh, where we require the target mesh as input and learn the progressive generation process. The intermediate figures illustrate the progressive results during auto-regressive generation. As shown, our method can accurately control geometric complexity; the number below our subfigure, such as 256, indicates that the mesh has exactly $256 + 1 = 257$ vertices. The face count can be indirectly and less accurately influenced by the number of vertices. We utilize the method described in Section 4.3 to learn the generation process using cross-entropy loss, as we essentially perform token classification after tokenization. The generation process always begins with a single vertex. We sincerely appreciate your interest in our implementation details!
>
> 6. **Any failure case examples?** Yes, we acknowledge that our method can sometimes yield unsatisfactory results, which we believe are primarily due to generalization issues. Unfortunately, due to this year's rebuttal policy, we are unable to include images here; instead, we would like to describe our findings.
> Each vertex split operation encodes both geometry (e.g., relative offset) and topology (e.g., the topological list), and we have observed that these failure cases reflect both aspects. Geometric artifacts may lead to local shifts; for instance, a chair's leg might be slightly offset by a translation vector of $(0.5, 0, 0)$ due to the discretization of the offset. While the overall topology remains unchanged, the leg is simply misplaced.
> Topological artifacts present more significant challenges. For example, a mesh may be generated correctly in the first 10 steps, but if an incorrect topology is predicted at the 11th step, this error can cause the result to deviate from the original intent, often resulting in an unrecognizable outcome as the shape falls out of distribution. We are committed to including a detailed analysis of these artifacts, complete with figure examples, in our revised paper. We sincerely appreciate your understanding and feedback as we work to improve our approach.

---

> > ### Comment · Reviewer_HBFZ · 2025-08-08
> >
> > Thank you for the detailed responses. It would be great to add the rebuttal response to the revision, in particular:
> >  - The four rules: it would be great to provide a rigorous proof in the supplementary material.
> >  - The details of the fitting experiments.
> >  - Failure cases (this also connects to the comments on errors from “overly flexible” topology): please make sure to add these discussions in the revision.
> >  - And improve the explanation of new concepts the algorithm details.

---

### Comment · Area_Chair_mgHa · 2025-08-02
**Reviewer-Author Discussions**

Hi Reviewers,

Thanks for your effort in reviewing for NeurIPS. We are now in the reviewer-author discussion phase. Please look at each others' reviews and the authors' responses, and further clarify any doubts, especially any points of disagreement with the authors before Aug 6 11:59pm AoE.

--AC

---

### Decision · Program_Chairs · 2025-09-17

**Decision:**

Accept (poster)

**Comment:**

This paper receives 2x accepts and 2x borderline accepts. The reviewers think that the proposed method is novel, the paper is well-written with clarity,  and the proposed method is technically sound and solid. The paper also shows good experimental results. Although there are some concerns on limited comparisons with other methods in the experiment section, the reviewers think that this is a minor concern and accept the paper. The ACs follow the ratings of the reviewers to accept the paper.